# Hate and Incivilities in Hashtags against Women Candidates in Chile (2021–2022)

Jarnishs Beltrán [1], Paula Walker [2] and René Jara [2,*]

[1] School of Informatics Engineering, University of Valparaíso, Valparaiso, IN 46383-6493, USA
[2] School of Journalism, University of Santiago de Chile, Ñuñoa, Santiago 7800020, Chile
* Correspondence: rene.jara@usach.cl; Tel.: +56-933835287

**Abstract:** This study is interested in the phenomenon of violence in social networks against women who hold political office in the framework of the Chilean constitutional process (2021–2022). To study this major socio-political phenomenon, we have used the tracking database "Women and Politics", composed of 2,912,732 Twitter posts mentioning women candidates and collected between July 2021 and September 2022. Based on this data, we analyzed the network of hashtags by electoral list at points in time: all 2021 messages, all 2022 messages and only hate messages published in 2022 (n = 563,223), codified by an automated detection method. The results of the study reveal that hashtags extracted from messages containing hate speech are better understood under the concept of incivilities. These were expressed in a two-phase electoral process: the presidential election and the constitutional plebiscite. The implications and limits of this study are discussed and considered in context in the conclusion.

**Keywords:** women; hate speech; online violence; constitutional process; Chile

## 1. Introduction

This paper addresses the violence published in social networks about women candidates, in the context of the Chilean Constitutional Process. After the social uprising of October 2019, a cross-party political agreement enabled a constitutional reform and plebiscite that allowed the people of Chile to approve the possibility of drafting a new constitution. In the context of a deep social and political crisis unprecedented in recent Chilean history, the country set out to find an institutional way to channel an acute social malaise that had accumulated over the past 30 years. The response to this problem was the organization of a Constitutional Convention, which incorporated gender criteria and quotas for indigenous peoples, as well as selecting the grouping of independent lists. The result was the election of a body with very heterogeneous and atypical members, with differing "political capitals" (Jofré-Rodríguez 2021), whose mission for one year was to draft a new constitutional proposal, which was put to a plebiscite on 4 September 2022.

The election of convention members took place on 15 and 16 May 2021 within the framework of unprecedented electoral rules. On an axis from left to right, the following lists were presented: Apruebo Dignidad, which grouped the candidates of the Partido Comunista de Chile and the parties of the Frente Amplio; Lista del Apruebo, grouping the center-left parties or Ex-Concertación; and Vamos por Chile, that grouped the alliance of the center-right parties RN, UDI and Evopoli. In addition to the lists, two other groupings were also presented: the one that brought together the candidacies of the Indigenous Peoples (Pueblos Originarios), who elected 17 representatives; and the list of *Independientes* candidates, who surprised by winning 38 seats, out of a total of 155.

The distribution of seats was also the product of an unprecedented parity rule, which ensured that half of the representatives were women. The new institutional rules had an clear effect on the expression of parity and also about indigenous peoples' representation, which makes it possible to study in detail the type of discourse circulating on social

networks about various groups excluded from society. In effect, the regulations approved (Law 21.216 2020) established that the process would ensure gender parity and with reserved seats for indigenous peoples, also allowing for the representation of lists outside the traditional party structures. Moreover, the gender parity of constituent power has meant a massive presence of women candidates across the country, with their public voice erupting in debates around contingent issues and others such as women's rights, feminism and discrimination (Figueroa 2021; Ponce de León 2021). The implementation of the measure was considered very successful, since parity was achieved and even in several cases, the rule operated in reverse: that is, some female candidates with very good electoral results had to give their seats to candidates who obtained only third place in the voting. Out of a total of 1373 nominations for the convention, 51% were women (699). In the end, seventy-seven women and 78 men were elected.

## 2. Theoretical Framework

### 2.1. Defining Hate Speech

We have preferred to use the notion of hate speech for this study, given that we have a panoramic view of the characteristics that the phenomenon assumed during the functioning of the Constitutional Convention in Chile. According to Parekh (2006), hate speech is: an objectively offensive or demeaning message; targeting a specifically identified social group and putting that group at risk of exclusion from society. Waldron (2012) expressed that hate speech manifests itself in four modalities: (a) accusing members of a specific group of committing illegal acts in a generalized manner; (b) equating that group with another element that allows its dehumanization; (c) denigrating and offensive characterization of the group; (d) specific prohibition according to the group's representative defining features. Starting from Miró's (2016) taxonomy, Calderón et al. (2020) identify three purposes pursued by these hate speeches: (1) direct incitement or glorification of violence; (2) incitement to discrimination, hate or restriction of rights; and (3) offenses against feelings.

### 2.2. Hashtags and Incivilities

Hashtags are one of the ways we have of approaching online communication phenomena that occur in social networks. From that point of view, hashtags represent a form of discrete and intentional expression of a message. In this particular case, our research focuses on studying hashtags extracted from a much larger corpus of tweets, which collected the total number of mentions of the women candidates. The study of hashtags has been frequently used to analyze feminist social movements (Mendes et al. 2019), social mobilizations (Suk et al. 2021) and, in general, public opinion phenomena such as the campaign #me too (Lindgren 2019). However, entry by hashtags has also been used to analyze hate speeches and, in particular, to study rational, ethnic and misogynist hate speeches circulating in social networks. In summary, it is possible to propose to formulate an analysis of hashtag networks for hate speeches against women in political office, provided that it is understood that the nature of the messages circulating under this type of mass messaging is not expressed under the same form and systematicity as in the cases of other hate speeches.

However, the use of hashtags seems to be less adapted to capture hate speech by itself, given that hashtags contain a message that seeks to circulate extensively. While individual tweets may contain more overt hate speech, this is not expected to be the case with hashtags. For this reason, we have considered it pertinent to use the notion of incivilities. A large part of the literature assumes that this violence is part of the nature of the exchanges that take place in social networks against politicians, and therefore observe this phenomenon through the prism of incivilities (Theocharis et al. 2020; Saldaña and Rosenberg 2020). By contrast, Gagliardone et al. (2015) define cyberhate as including expressions that directly encourage the commission of discriminatory acts or hate violence. This conc9eptualization leads Wright et al. (2021, p. 22) to say that "it is a central and highly relevant scientific and social issue", while the newly termed concept of cyberhate requires its own field of study

(Chakraborti et al. 2014). Finally, Davidson et al. (2017) proposed distinguishing between offensive or vulgar language and hate speech.

The use of the concept of incivilities is further justified by the issues in which this research is framed: women's participation in political institutions and the discourses that circulate about them in social networks. Therefore, this work is part of the line of research that is concerned with the gender gap in the field of politics. Among these concerns, one that is fundamental is the violence that women who perform representative functions receive via social networks. Indeed, several studies have shown that these people are more exposed to this type of violence due to high levels of public recognition (Rheault et al. 2019, p. 1; Krook and Restrepo 2020; Southern and Harmer 2021; Suarez Estrada 2021). By posing the question of how hostile behavior against women affects the Chilean constitutional process, we will gain a much better understanding of the challenges they face in consolidating their careers, particularly in an international context of an acute and diverse crisis of democracy.

In order to answer this major question, we re-constructed the networks of 50 hashtags with the highest circulation for the four main and one additional minor electoral lists that ran for constitutional elections (five in total: Independientes, Pueblos Indígenas, Apruebo Dignidad, Lista del Apruebo and Chile Vamos). In this way, a comparison was made between the total hashtags mentioning the members of each list for the period 2021 and 2022. That same information was compared with the hashtags that were extracted from the messages that were effectively coded as hate speech for the year 2022, by means of the automatic detection method that we describe below. From this network analysis exercise that emerged from the analysis and its metrics, we explored whether there were substantive differences between the hashtags that appeared most frequently in the different electoral lists (RQ 1) and whether there were detectable differences in each electoral list for the three observations (2021 without hate/2022 without hate/2022 hashtags with hate) (RQ 2).

The analysis is presented in two sections: (a) the hashtags and their metrics are presented for each list, and (b) the network graphs of all lists for the three data corpora (2021 total hashtags, 2022 total hashtags and 2022 hashtags with hate messages) are then introduced.

## 3. Methods

### 3.1. Sample and Procedure

We used a quantitative methodology in two phases. The first phase, based on natural language processing techniques, had as its main objective the detection of these discourses. With the support of the Stop Hate project of the Audiovisual Content Observatory of the University of Salamanca, Spain, we executed the download of tweets using Python through Twitter's Application Programming Interface 2 (API). The search was carried out using the usernames of the different members without the 'at' symbol (@) to access both their tweets and those that mentioned them. All these messages and their associated microdata were downloaded—language, geolocation (if any), public tweet metrics, and public metrics of the user, among others.

The second stage consisted of the characterization of these speeches, based on manual coding. A total of 2,912,732 tweets were downloaded between July 2021 and September 2022. Table 1 below describes the distribution of the data collected by each discharge month (month of the year) and by the list that represented each woman candidate during her election (electoral lists).

**Table 1.** General corpus of Twitter messages by electoral list, 2021–2022.

| Date | Lista del Apruebo | Apruebo Dignidad | Independientes | Pueblos Originarios | Vamos por Chile | Total |
|---|---|---|---|---|---|---|
| July-21/December-21 | 50,464 | 237,360 | 308,059 | 487,220 | 624,343 | 1,707,446 |
| January-22–September-22 | 40,705 | 214,513 | 329,439 | 131,913 | 676,520 | 1,393,090 |
| Total | 84,190 | 428,404 | 591,286 | 602,441 | 1,206,411 | 2,912,732 |

As can be seen in the table, the candidates are not distributed evenly across the different lists. Some lists, such as "Vamos Chile", returned far larger numbers of mentions than others. This is not related to the number of representatives that each list elected: For example, in the second biggest list of mentions (from "Pueblos Originarios"), the candidates in question represent only 11 of the total seats, including candidates of both sexes. Finally, the "Lista del Apruebo" obtained a very small number, and in this case, a good part of its candidates are mentioned. All this means that it was not the number of representatives that explains the number of mentions, but rather the notoriety and/or interest that the candidates caused in Twitter users.

From this general corpus, a subcorpus of messages was extracted which, according to the detection method used, contained hate speech. The distribution by list and month is shown in the Table 2. As can be seen, all the lists studied record hate speech in their messages, showing the political transversality of the phenomenon. A second aspect of interest is that the number of hate messages detected is proportional to the total number of messages that mention the candidates on each list. It should be noted that the dictionary used for the automatic detection went from 102 words to 5379 words, thanks to the improved detection and the adaptations to the use of Chilean Spanish that the project team was able to identify.

**Table 2.** Subcorpus of Twitter messages with hate speech detected by electoral lists (2022).

| Month/Electoral Lists | Lista del Apruebo | Apruebo Dignidad | Independientes | Pueblos Originarios | Vamos por Chile | Total |
|---|---|---|---|---|---|---|
| January | 2499 | 4397 | 11,089 | 7156 | 8755 | 33,896 |
| February | 4968 | 8749 | 19,640 | 7577 | 42,916 | 83,850 |
| March | 3755 | 8918 | 17,673 | 8980 | 35,466 | 74,792 |
| April | 3900 | 17,698 | 26,354 | 13,732 | 44,795 | 106,479 |
| May | 1097 | 6044 | 4261 | 7617 | 7198 | 26,217 |
| June | 1163 | 8569 | 17,880 | 6827 | 25,749 | 60,188 |
| July | 1184 | 10,046 | 17,479 | 7096 | 51,489 | 87,294 |
| August | 8749 | 17,352 | 14,679 | 3415 | 38,071 | 82,266 |
| September | 635 | 1341 | 2000 | 1013 | 3252 | 8241 |
| Total | 27,950 | 83,114 | 131,055 | 63,413 | 257,691 | 563,223 |

*3.2. Measures*

3.2.1. Electoral Lists

In view of the significant scores obtained by the various groups calling themselves "independents", we have grouped them into a single category. However, this exercise is not without its problems. For this choice, it is possible to identify at least two types of independents—independents within a given list or pact, i.e., those persons who were not party militants but were sponsored by the parties to integrate their lists; and the lists of independents. According to the analysis of Rozas Bugueño et al. (2022), of the 155 elected candidates, at least 103 did not belong to a political party (p. 80). Of these 103, 55 can be considered non-partisan independents, that is, candidates not supported by parties and without militancy, and of these, 33 were women (p. 82). However, it is possible to point out that from an ideological point of view, the label of independent does not fully reveal in name the complexity of the variable considered for the analysis. Indeed, as Fábrega (2022) has

pointed out, the ideological identification of each candidate prior to the process provided a variable that predicted very well their legislative behavior within the Convention. Therefore, it is possible to point out that in some cases, the category of independents is too large to perceive, for example, the discrepancies that existed in the positions defended by the members of Lista del Pueblo and those of the list of Independientes No Neutrales. Such is the heterogeneity of the lists that participated and of the possible combinations, that we have preferred to group under the category "Independent" all those candidates who belonged to this complex series of lists without belonging to the traditional political parties in Chile.

### 3.2.2. Hashtags' Network Metrics and Visualization

We performed the metric analysis of hashtags extracted from the Twitter messages posted. We used an undirected graph network G to represent the connections between the fifty most frequent hashtags in tweets. In G = (V, E), V denotes the set of nodes (hashtags) and E denotes the set of edges (co-occurrence of two hashtags in the same tweet) in G. An edge $e_{ij} \epsilon$ E corresponds to a set of node pairs $(v_i, v_j)$ that connects node $v_i$ and $v_j$ in G. Edges in the network were defined after Kang et al. (2017) as follows.

Degree centrality simply corresponds to the degree of a node, i.e., the number of edges that a node has with the other nodes. Closeness centrality is based on calculating the average of the geodesic distances (or shortest paths) from one node to all other nodes. The greater the distance between two vertices, the smaller the closeness between them. Therefore, closeness is defined as the multiplicative inverse of the remoteness between two vertices. Betweenness centrality is a measure of centrality that quantifies the number of times a node lies between the geodesics of other nodes. A node will have high betweenness if it is a cutoff vertex for many geodesics between nodes (Drieger 2013; Gloor and Diesner 2014; Kang et al. 2017; Mattei et al. 2021). To detect communities, we used Community detection based on edge betweenness. High-betweenness edges are removed sequentially (recalculating at each step) and the best partitioning of the network is selected (Girvan and Newman 2002). For the construction of the network and the calculation of the metrics, we used the package igraph (Csardi and Nepusz 2006). Figure 1 summarizes the three steps required, which are as follows.

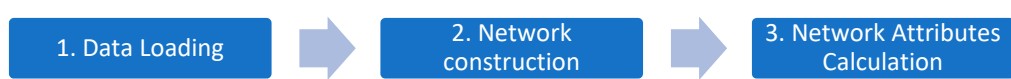

**Figure 1.** Description of network construction stages.

1.  Loading the data. In this step, the data previously labeled as tweets with "hate speech" loading are read.
2.  Network construction. In this step, each tweet is considered as a document. A process of separating by "token" is performed obtaining a matrix of 0 and 1—0 if the token is not in the tweet and 1 if the token is in the tweet. Then, only the hashtags in the document array are selected prior to creating a "sparse feature co-occurrence matrix" (Benoit et al. 2018).
3.  Network attributes calculation. In this step, the overall network metrics are calculated: density, diameter and average path length. Moreover, the local centrality measures (degree centrality, betweenness centrality and closeness centrality) are calculated.

### 3.2.3. Detecting Hate Speech and Incivilities

To detect messages containing hate speech and incivilities, we used a double coding process, first manual (2021) and then automated. First, the predictive model classified the messages into hate/non-hate speech (first coding) and second, this classification was manually validated by a researcher (second coding). In a first stage, the intercoder reliability index between the three manual coders was controlled, with a result of 0.70 for the June 2021 data, 0.79 for that of July 2021 and 0.80 for the August 2021 data. Next,

the intercoding agreement between manual and automated coding was checked. Thus, a 0.50 manual intercoding agreement was obtained for September and October 2021 and 0.68 for November and December 2021. These results, although lower, are still within the commonly used standards for this type of automated detection procedure.

To detect incivilities, we operated a qualitative manual analysis, from the five most frequent hashtags up to the total network of 50 hashtags. Emphasis was placed on the dimension of offensive, personal discrediting and vulgar expressions that can be summarized in hashtags. We read each hashtag and, when there were doubts about their meaning, we went back to check in the database the meaning that this expression may have had for the accounts that used them at the time.

## 4. Results

We organized the analysis of the hashtags into two points in time: 2021 and 2022. Here, we present only the metrics that the networks collected, highlighting the contrasts seen in the topics of each network. The first point that seems important is the analysis of the hashtag networks metrics. The observation is given for each year separately, thus favoring the analysis of the possible differences that each network articulates. Secondly, we show the metrics of the hashtag's networks. Tables 3–8 show the five hashtags with the highest metrics of the five electoral and studied lists. Subsequently, in Figures 2–6, we present the graphs of each network, which allow the hashtags containing incivilities to be analyzed. An interpretation of the totality (50) of the hashtags of the network is given.

**Table 3.** Five highest hashtags in local network metrics for "Apruebo Dignidad".

|  | Hashtags | Degree Centrality | Hashtags | Betweenness Centrality | Hashtags | Closeness Centrality |
|---|---|---|---|---|---|---|
| 2021 | #convencionconstitucional | 0.058234098 | #chile | 0.061760646 | #chile | 0.0003016006 |
|  | #chile | 0.055910543 | #convencionconstitucional | 0.057163137 | #convencionconstitucional | 0.0003015991 |
|  | #boricpresidente | 0.042550102 | #boricpresidente | 0.031703071 | #boricpresidente | 0.0003015927 |
|  | #convencionconstituyente | 0.028173105 | #convencionconstituyente | 0.020692143 | #convencionconstituyente | 0.0003015906 |
|  | #boricpresidente2022 | 0.026575661 | #circoconstituyente | 0.016701160 | #convenciónconstitucional | 0.0003015882 |
| 2022 | #apruebo | 0.062204046 | #apruebo | 0.043150557 | #rechazo | 0.0004489131 |
|  | #rechazo | 0.061127852 | #rechazo | 0.042513213 | #chile | 0.0004489101 |
|  | #chile | 0.048428756 | #chile | 0.035170520 | #apruebo | 0.0004489071 |
|  | #rechazotransversal | 0.041541111 | #convencionconstitucional | 0.025602884 | #convencionconstitucional | 0.0004489018 |
|  | #convencionconstitucional | 0.036375377 | #nuevaconstitucion | 0.020994198 | #rechazotransversal | 0.0004488966 |

**Table 4.** Five highest hashtags in local network metrics for "Lista del Apruebo".

|  | Hashtags | Degree Centrality | Hashtags | Betweenness Centrality | Hashtags | Closeness Centrality |
|---|---|---|---|---|---|---|
| 2021 | #convencionconstitucional | 0.057928613 | #convencionconstitucional | 0.0470311662 | #convencionconstitucional | 0.001036760 |
|  | #chile | 0.047981276 | #chile | 0.0304784154 | #chile | 0.001036705 |
|  | #circoconstituyente | 0.047981276 | #circoconstituyente | 0.0272174545 | #rechazodesalida | 0.001036678 |
|  | #rechazodesalida | 0.039204213 | #maluchapinto | 0.0260586534 | #circoconstituyente | 0.001036676 |
|  | #maluchapinto | 0.037448800 | #rechazodesalida | 0.0193613223 | #teleton | 0.001036632 |
| 2022 | #rechazo | 0.087579618 | #rechazo | 0.0466066455 | #rechazotransversal | 0.001656769 |
|  | #rechazotransversal | 0.070063694 | #rechazotransversal | 0.0405721383 | #rechazo | 0.001656677 |
|  | #rechazoelmamarrachocomunista | 0.064490446 | #convencionconstitucional | 0.0379152858 | #rechazoladestrucciondechile | 0.001656634 |
|  | #convencionculia | 0.056528662 | #rechazodesalida | 0.0279228956 | #rechazoelmamarrachocomunista | 0.001656618 |
|  | #convencionconstitucional | 0.050955414 | #convencionculia | 0.0278270625 | #convencionculia | 0.001656542 |

**Table 5.** Five highest hashtags in local network metrics for "Independientes".

|  | Hashtags | Degree Centrality | Hashtags | Betweenness Centrality | Hashtags | Closeness Centrality |
|---|---|---|---|---|---|---|
| 2021 | #chile | 0.088442087 | #chile | 0.082362316 | #chile | 0.0002921714 |
|  | #convencionconstitucional | 0.068829407 | #convencionconstitucional | 0.062670888 | #convencionconstitucional | 0.0002921693 |
|  | #tiapikachu | 0.047119773 | #tiapikachu | 0.032845669 | #tiapikachu | 0.0002921613 |
|  | #antofagasta | 0.040582213 | #antofagasta | 0.026173946 | #convencionconstituyente | 0.0002921604 |
|  | #circoconstituyente | 0.035031454 | #circoconstituyente | 0.024310114 | #antofagasta | 0.0002921602 |
| 2022 | #chile | 0.062228871 | #chile | 0.0460340376 | #chile | 0.0003298569 |
|  | #rechazo | 0.060882573 | #convencionconstitucional | 0.0431648988 | #rechazo | 0.0003298563 |
|  | #rechazotransversal | 0.057142857 | #rechazo | 0.0364276174 | #convencionconstitucional | 0.0003298548 |
|  | #convencionconstitucional | 0.052505610 | #apruebo | 0.0277960151 | #rechazotransversal | 0.0003298537 |
|  | #apruebo | 0.048017951 | #nuevaconstitucion | 0.0235721928 | #apruebo | 0.0003298518 |

**Table 6.** Five highest hashtags in local network metrics for "Pueblos Originarios".

|  | Hashtags | Degree Centrality | Hashtags | Betweenness Centrality | Hashtags | Closeness Centrality |
|---|---|---|---|---|---|---|
| 2021 | #elisaloncon | 0.107892527 | #elisaloncon | 0.0638686641 | #elisaloncon | 0.0001674043 |
|  | #chile | 0.090330255 | #convencionconstitucional | 0.0539812099 | #convencionconstitucional | 0.0001674038 |
|  | #convencionconstitucional | 0.087881332 | #chile | 0.0538189096 | #chile | 0.0001674037 |
|  | #circoconstituyente | 0.067450322 | #circoconstituyente | 0.0431164686 | #circoconstituyente | 0.0001674023 |
|  | #convencionconstituyente | 0.062902323 | #convencionconstituyente | 0.0314237062 | #convencionconstituyente | 0.0001674023 |
| 2022 | #elisaloncon | 0.093133386 | #elisaloncon | 0.0563966388 | #elisaloncon | 0.0005901172 |
|  | #chile | 0.076295712 | #chile | 0.0484200257 | #chile | 0.0005901166 |
|  | #rechazo | 0.073138648 | #rechazo | 0.0377312987 | #rechazo | 0.0005901082 |
|  | #rechazotransversal | 0.056300973 | #convencionconstitucional | 0.0365022372 | #rechazotransversal | 0.0005900961 |
|  | #convencionconstitucional | 0.052091555 | #apruebo | 0.0250988344 | #convencionconstitucional | 0.0005900933 |

**Table 7.** Five highest hashtags in local network metrics for "Vamos por Chile".

|  | Hashtags | Degree Centrality | Hashtags | Betweenness Centrality | Hashtags | Closeness Centrality |
|---|---|---|---|---|---|---|
| 2021 | #convencionconstitucional | 0.057928613 | #convencionconstitucional | 0.0470311662 | #convencionconstitucional | 0.001036760 |
|  | #chile | 0.047981276 | #chile | 0.0304784154 | #chile | 0.001036705 |
|  | #circoconstituyente | 0.047981276 | #circoconstituyente | 0.0272174545 | #rechazodesalida | 0.001036678 |
|  | #rechazodesalida | 0.039204213 | #maluchapinto | 0.0260586534 | #circoconstituyente | 0.001036676 |
|  | #maluchapinto | 0.037448800 | #rechazodesalida | 0.0193613223 | #teleton | 0.001036632 |
| 2022 | #rechazo | 0.075341111 | #rechazo | 0.0442381727 | #rechazo | 0.0002013888 |
|  | #chile | 0.064563971 | #chile | 0.0426955224 | #chile | 0.0002013877 |
|  | #rechazotransversal | 0.048348823 | #apruebo | 0.0219553850 | #rechazotransversal | 0.0002013857 |
|  | #apruebo | 0.043602927 | #rechazotransversal | 0.0179918509 | #rechazodesalida | 0.0002013851 |
|  | #convencionculia | 0.033814515 | #convencionculia | 0.0144377625 | #apruebo | 0.0002013850 |

**Table 8.** Five highest hashtags in local network metrics with hate speech (2022).

|  | Hashtags | Degree Centrality | Hashtags | Betweenness Centrality | Hashtags | Closeness Centrality |
|---|---|---|---|---|---|---|
| Apruebo Dignidad | #rechazo | 0.093913744 | #rechazo | 0.0722973939 | #rechazo | 0.0009527240 |
|  | #rechazotransversal | 0.060056429 | #chile | 0.0455788406 | #chile | 0.0009526622 |
|  | #apruebo | 0.056831923 | #apruebo | 0.0385706469 | #rechazoelmamarrachocomunista | 0.0009526479 |
|  | #chile | 0.054816606 | #convencionculia | 0.0325274655 | #rechazotransversal | 0.0009526464 |
|  | #rechazoelmamarrachocomunista | 0.054413543 | #rechazotransversal | 0.0321835020 | #convencionculia |  |
| Lista del Apruebo | #rechazo | 0.09129512 | #rechazo | 0.0577913241 | #rechazotransversal | 0.002306037 |
|  | #rechazoelmamarrachocomunista | 0.07112527 | #convencionconstitucional | 0.0459548109 | #rechazo | 0.002305947 |
|  | #rechazotransversal | 0.07112527 | #rechazotransversal | 0.0425674794 | #rechazodesalida | 0.002305806 |
|  | #convencionconstitucional | 0.05944798 | #rechazodesalida | 0.0374297950 | #rechazoelmamarrachocomunista | 0.002305698 |
|  | #convencionculia | 0.05307856 | #convencionculia | 0.0341646357 | #convencionculia | 0.002305687 |
| Independientes | #rechazo | 0.083732057 | #rechazo | 0.054970093 | #rechazo | 0.0007381898 |
|  | #rechazodesalida2022 | 0.064892344 | #convencionconstitucional | 0.046320314 | #chile | 0.0007381690 |
|  | #rechazotransversal | 0.064294258 | #chile | 0.045161088 | #rechazodesalida2022 | 0.0007381662 |
|  | #chile | 0.061004785 | #rechazodesalida2022 | 0.032766792 | #rechazotransversal | 0.0007381652 |
|  | #convencionculia | 0.061004785 |  |  | #convencionconstitucional | 0.0007381581 |

**Table 8.** *Cont.*

| | Hashtags | Degree Centrality | Hashtags | Betweenness Centrality | Hashtags | Closeness Centrality |
|---|---|---|---|---|---|---|
| Pueblos Originarios | #rechazo | 0.094594595 | #rechazo | 0.051852527 | #rechazo | 0.001107906 |
| | #elisaloncon | 0.081547064 | #elisaloncon | 0.045490531 | #rechazodesalida2022 | 0.001107885 |
| | #rechazodesalida2022 | 0.076421249 | #chile | 0.039661442 | #elisaloncon | 0.001107885 |
| | #rechazotransversal | 0.072693383 | #convencionconstitucional | 0.039510053 | #rechazotransversal | 0.001107876 |
| | #chile | 0.068965517 | #rechazodesalida2022 | 0.039278551 | #chile | 0.001107868 |
| Chile Vamos | #rechazo | 0.094194962 | #rechazo | 0.063475465 | #rechazo | 0.0004214846 |
| | #chile | 0.074297189 | #chile | 0.051056471 | #chile | 0.0004214772 |
| | #apruebo | 0.054581964 | #apruebo | 0.027644181 | #rechazotransversal | 0.0004214643 |
| | #rechazotransversal | 0.051113545 | #convencionculia | 0.023711760 | #rechazodesalida | 0.0004214632 |
| | #convencionculia | 0.048010223 | #rechazotransversal | 0.017433542 | #rechazoporchile | 0.0004214626 |

## 4.1. Highest Hashtags Networks Local Metrics by Electoral Lists (2021–2022)

The representation of these networks is shown in Figures 2–6 below. Firstly, we have in Figure 2 the representation of the network of hashtags of the *Apruebo Dignidad* membership. This list represented the left-wing sensibilities of the political spectrum, which eventually integrated with the government at the time of the election in December 2021 of President Gabriel Boric Font. In this network, in 2021, the hashtags refer mostly to the constitutional process, giving way also to messages that make clear reference to the then candidate Gabriel Boric. There are also some references to the right-wing candidate José Kast and to rejection, but these are not central to the network. However, in 2022, we find a complete turnaround in the mentions of the members. The hashtags are oriented in this second phase towards the plebiscite process, with the call to approve the process clearly appearing (#apruebo). At the same time, the discourse of rejection also appears very strongly, with various hashtags showing that mentions of the members are strongly associated with this discourse (#rechazodesalida, #circoconstituyente or #rechazoelmamarracho).

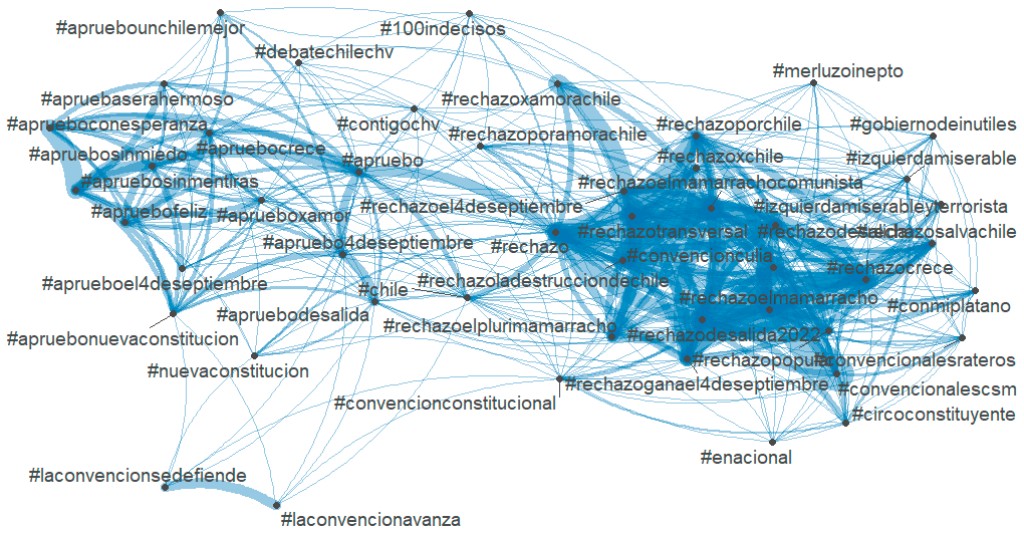

**Figure 2.** Network of hashtags with hate speech for *Apruebo Dignidad 2022*.

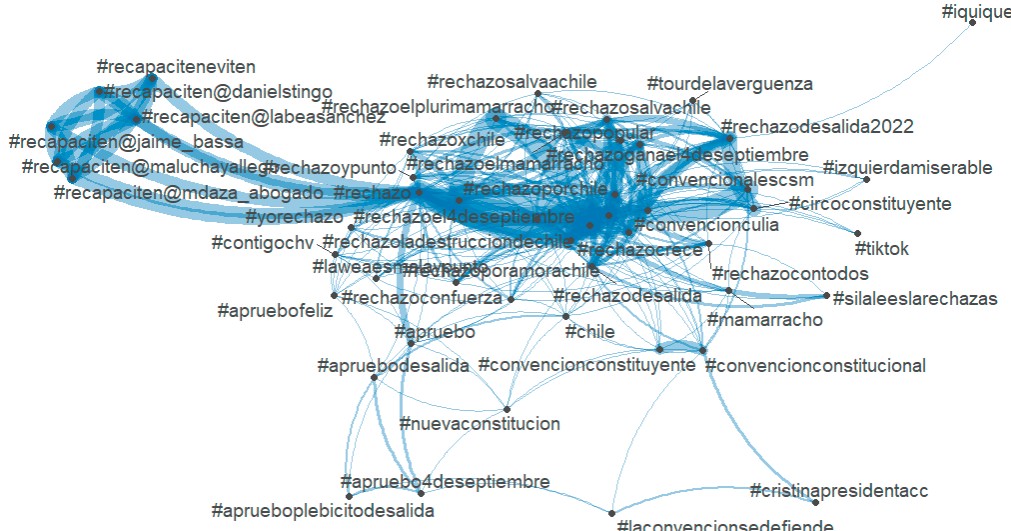

**Figure 3.** Network of hashtags with hate speech for *Lista del Apruebo* 2022.

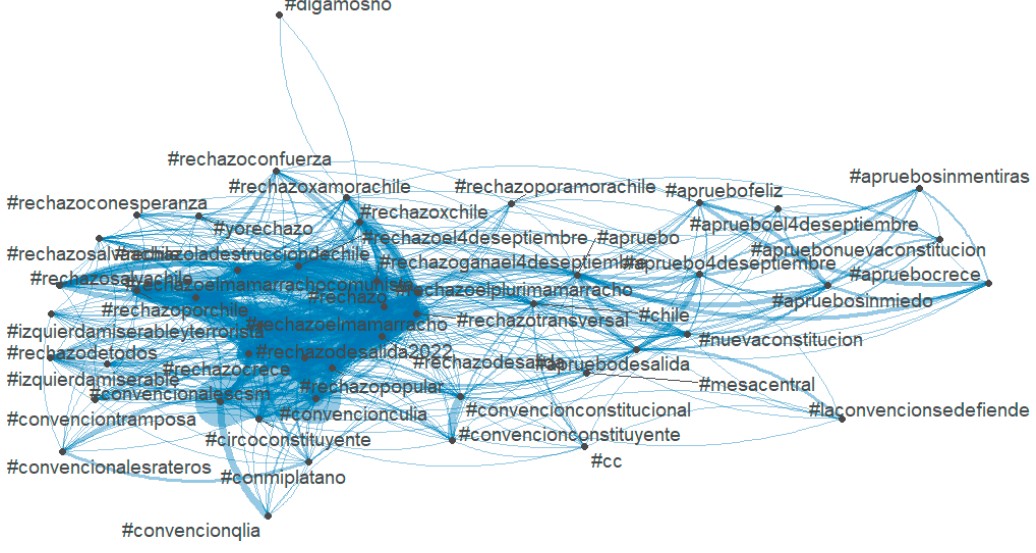

**Figure 4.** Network of hashtags with hate speech for *Independientes* 2022.

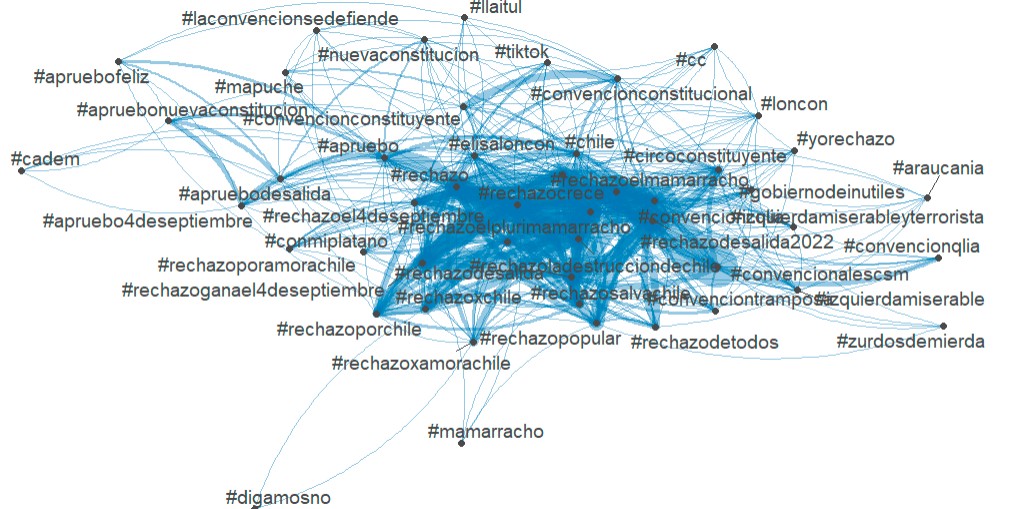

**Figure 5.** Network of hashtags with hate speech for *Pueblos Originarios* 2022.

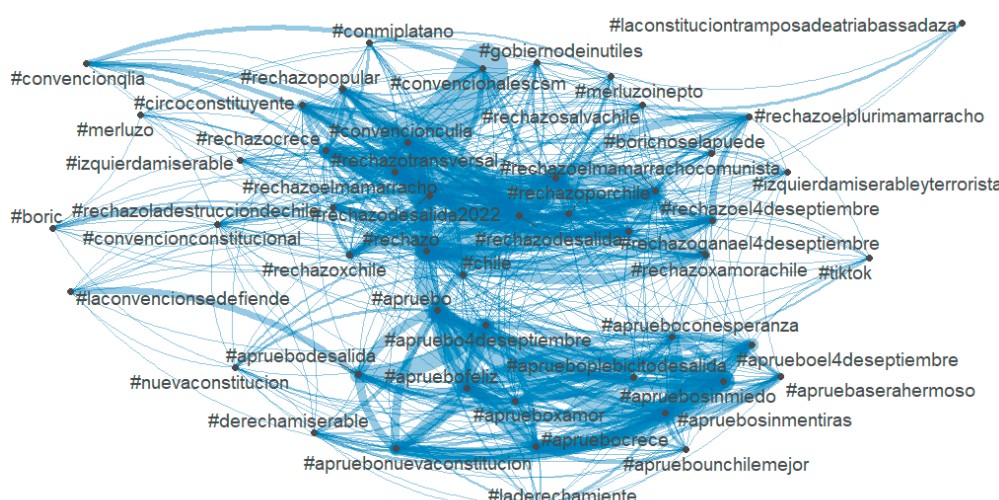

**Figure 6.** Network of hashtags with hate speech for *Vamos por Chile* 2022.

Overall, in the network "Apruebo Dignidad 2021", we find that the conversation focused on the figure of the current president, where #boricpresidente stands out with a between centrality of 99.6, thus occupying third place in this metric, and is the first hashtag that is not directly related to the process. In terms of communities, there are two communities with 28 16 nodes, respectively. The first community employs an emblem that exalts the convention, while that of the second one is the figure of the current president, who was an opposition candidate to the current government. In the "Apruebo Dignidad 2022" network, it is observed that the density of the network is double that of 2021, which implies greater connectivity between them. In addition, it is observed that the centrality metrics in favor of rejection and approval are practically the same. For this aspect, we find two communities of 27 and 12 nodes, respectively; however, unlike the 2021 network, the conversation in the larger community revolves around the rejection of the new constitution project.

In the network "Lista del Apruebo 2021" and "Lista del Apruebo 2022", four of the five hashtags with the highest degree of centrality promote the rejection of the constitutional process, which implies that these hashtags were recurrently in the conversations mentioning the members of the "Lista del Apruebo". Meanwhile, the betweenness centrality for the "Lista del Apruebo 2022" network shows very high #rejection, implying that this hashtag was bridging many conversations. Looking at the communities, we can see both networks have a large community in which the terms referring to being against the constitutional process are frequently used.

In both networks, the largest community revolved around voting against the project and against the figure of the convention's president in the first period, namely, Elisa Loncon.

A specific case is represented by the networks of hashtags that we were able to construct from the mentions of women candidates who were part of the lists of "Independientes". It is known that the possibility of integrating lists of independents, as well as gender parity, was one of the main innovations of this process: it is therefore interesting to scrutinize these data in detail, since the ideas of these lists do not represent either the thinking or the ideas of the two previous lists, traditionally represented in Congress since the return to democracy. It is perhaps for this reason that the network of hashtags of this list represented in Table 4 expresses a wide dispersion of themes and discourses, ranging from the defense and vindication of the legitimacy of the convention, to the particular mention of specific candidates. There are also some allusions to the presidential candidate Boric and to specific issues discussed at the convention (#semanaterritorial and #plebiscitosdirimentes). In the second phase, the mentions in the hashtags network become monothematic, with the by far most significant issue discussed being the option against the plebiscite, which appears in the most varied modulations (#rechazosalvaachile, #rechazoconfuerza,

#recha-zoel4deseptiembre, #noesmiconstitucion, #rechazoxamorachile). Rejection is then observed as a much denser and interconnected group of mentions at the top of the network, while the approval node is observed as less dense and with fewer nodes than the rejection network.

In the "Independientes 2021" network, the conversation focuses on the convention and its work in the regions of the country; and the largest of its ten groups mentions the convention and its actors only. In the "Independientes 2022" network, the centrality metrics indicate that the conversation goes from being neutral to a conversation charged against the process, thus polarizing towards one of the sectors. We found ten communities, the largest of which having 21 nodes, where the conversation is centered against the convention.

It seems interesting to analyze the network of hashtags that mention women candidates entering the convention in the quotas set aside for indigenous peoples. We know from previous descriptive studies that violence against women candidates was more intense in this subgroup, so that in this case it is possible to point out that it is not only about incivilities, but also about hate speech, which mixes attacks based on gender, race and social class. In this case, Table 6 shows a clear centrality in the first phase in 2021 of the constitutional convention process, with the figure of Elisa Loncón standing out as the main representative of this space. The sub-network articulated in the upper part of the figure, which integrates criticisms of the policy of pension fund withdrawals and the work of the convention, is a curious matter. In contrast to the first, the second phase shows a clear predominance of rejection in its different modalities (#re-chazotransversal, #rechadodesalida, #rechazocrece, #rechazoporchile), together with critical allusions to Boric's government. Mentions of the 'Apruebo' option are marginal and not very dense.

In the networks "Pueblos Originarios 2021" and "Pueblos Originarios 2022", the centrality metrics place #rechazo in the second position, after #chile, being a direct allusion to being against the constitutional process. In addition, the other hashtags also call to vote against the project of the new constitution. It is important to mention that in the network "Pueblos Originarios 2021", the hashtag #elisaloncon is in fifth place in the centrality metrics, which led us to think about the existence of messages loaded with rejection towards the figure of this candidate.

When we review the hashtags where candidates from the "Vamos por Chile" lists are mentioned, we can see in the first phase of 2021 a fairly dense graphic with many connections. The members of these collectives, who represented the parties and sensibilities of the Chilean right, were an important voice within the convention. Despite the fact that the representation of these collectives was minimal in number within the convention, they managed to make their presence felt, and for this very reason were on the receiving end of a considerable amount of hate speech during the period under study. The references refer to contingent public policy and legislative issues such as pension fund withdrawals, as well as explicit references to the presidential candidates, Kast and Boric. Mention is also made to some candidates in particular (Elisa Loncon or Teresa Marinovic) and to the convention in general (#constituyentesflaites). References to rejection are still very limited, with the scenario of the 2021 presidential run-off in Chile monopolizing the discussion.

The tendency to orient hashtags towards rejection is shown in the majority in the second phase in 2022. The references articulate a series of ways to justify rejection in the exit plebiscite of 4 September 2022, which itself seem to be the main justification for mentioning the specific candidates. Marginally, there are hashtags that direct their criticism against elements of contingent politics, and against the performance of President Boric and his ministers (#renunciasiches). The central hashtags of the network are approval and rejection, both articulating different forms and modalities of agreement and rejection of the constitutional reform proposed.

In the "Vamos por Chile 2021" network, the centrality metrics show that the nodes with the highest scores pointed to the discrediting of the process and called for the rejection option. In addition, we found two distinct communities: the conversation of the largest one revolved around the discrediting of the convention and its president; the other community

points to the discrediting of the ruling pact of the time and of those who were from the same political current as this list. In the network "Vamos por Chile 2022" the centrality metrics show that the discourse focuses only on the rejection option. Ten communities were found, the largest with 29 nodes, whose conversation encourages the choice of the option to reject the new constitution.

### 4.2. From Hate Speech to Incivilities

In a second stage of the analysis, we analyzed the metrics of the network of hashtags with messages in which we detected the presence of hate speech. We proceeded in the same way as with that information downloaded for the network of mentions for the year 2021 and 2022. However, in this section we present the hashtags and metrics for all the lists gathered. The result of this overview is seen in Table 8.

The information presented in the table follows a clear pattern. Of the 75 hashtags recorded (5 possible for each metric), there is a high recurrence of themes. Indeed, 42 of them make mention of the rejection of different modalities, while 14 hashtags include the word Chile, showing the importance that the question of identity and nationality has gained in this debate. Hashtags mentioning the convention numbered 13, those that did so for the approval option were 6 in number, and 3 include the name of the President of the commission, Elisa Loncón.

These results allow us to verify that there is an evident coincidence between the presence of hate speech in tweets and the presence of messages related to the campaign of the rejection of the new constitution. This situation is transversal to the hashtag's networks of the different electoral lists, which shows the important presence of this discourse in most of the discussions that circulated on Twitter. The centrality of the rejection discourse in these networks is very eloquent.

Other results that can be extracted from the comparison between lists is the cross presence of antagonistic discourses in these networks. Approval is more present in the hashtags of *Apruebo Dignidad* and *Lista del Apruebo*, but also in the *Vamos por Chile*'s network. Conversely, the rejection network is very strong in the first two former lists, which indicates that the mentions of candidates with hate speeches were effectively crossed: those who were for the approval mentioned candidates of the lists that were for the rejection; conversely, those who were for the rejection made obvious mention in their hate tweets of candidates who were for the approval.

Beyond the five hashtags with the highest metrics, we can go deeper into the political analysis of each network thanks to their network visualizations. The following figures plot the position of the 50 hashtags that comprise them. The graphs for the *Lista del Apruebo* and *Apruebo Dignidad* lists are shown in Figures 2 and 3.

In the network of hashtags of *Apruebo Dignidad*, rejection is not only present, but represents the main subnetwork. A darker core is clearly observed in Figure 2 on the right side of the image. This implies that there are many links between the hashtags, which makes the network denser in that sector. Conversely, the presence of approval is plotted on the right side of the image, by several nodes but with less density.

In the network of hashtags of the *Lista del Apruebo* representatives, the presence of messages alluding to rejection is central but less dense than in the *Apruebo Dignidad* network, for example. The allusions to approval are found in the lower part of the graph, in a dispersed and not very dense manner. An interesting aspect that appears in this network is a nucleus with hashtags that formulate the concept "recapacitar" (please reconsider) in various ways. These hashtags integrate the names of women, alluding to their extreme positions they took within the Convention. Other hashtags that appear and that could be qualified as incivilities are #rechazosalvarachile, #izquierdamiserable, #rechazoelplurimamarracho, #la-weaesmalaypunto.

In the case of the *Independientes* network of hashtags, a denser nucleus linked to rejection and its different modalities can be observed in the low part of the network, on the left side. It has in its center two hashtags alluding to the convention: #rechazoelmamarracho

and #rechazoelplurimamarracho. Many of the hashtags do not seem to carry negative messages, but rather convey positive ideas associated with rejection, for example: #rechazodetodos, #rechazoconesperanza, #rechazopopular, #rechazoconesperanza. Contrary to what is observed with rejection, allusions to approval are marginal. Among the negative messages, there are mentions of the regulations approved by the convention (#convencionalesrateros) and to the convention itself (#convenciontramposa, #convencionqlia).

In the case of the network of hashtags for the *Pueblos Originarios* list (Figure 5), the approval option is hardly mentioned. As in the other cases, the messages alluding to the rejection option are in the majority. The hashtag #elisaloncon is central to this network: this is logical in view of her public significance, yet also shows the permanent vigilance that surrounded her as a public figurehead of the convention, and which effectively besieged her. It is interesting to note that there are several hashtags that clearly contain incivilities and insults showing the negative mood with which Twitter users expressed themselves against the women representatives of this group. Some examples were: #mamarracho, #izquierdamiserableyterrorista, #zurdosdemierda, #convencionqlia, #convencioncsm. Finally, it is worth noting the allusion to the Mapuche conflict with the State of Chile (#llaitul, #mapuche, #araucanía), which shows how the performance of this list was related to the development of this conflict in the southern part of the country.

In the case of the *Vamos por Chile* network (Figure 6), it can be observed that mentions are evenly balanced between hashtags alluding to approval and rejection, with a core of hashtags clearly identified with approval in the lower part of the network. In the upper part, we clearly find the presence of hashtags linked to the rejection option. In the middle of both we find the hashtag #chile, which seems to unite and connect both networks, since it is in the center of both cores.

## 5. Discussion of Results and Conclusions

How can we understand the nature of the violence suffered by women in politics? Social networks have only increased this phenomenon, to the extent that they have become a sounding board for hate speech and incivilities that target certain groups in society. In this way, the social and political context in Chile during the constitutional process has offered an important opportunity to study the underlying mechanisms of this phenomenon.

The results show, in line with the literature, that most of the interactions with women politicians' on their Twitter accounts contain violent speeches of different nature. The analysis not of the messages themselves but of the hashtag networks is particularly practical for revealing the political interests and macro-discourses from which violence is perpetrated against women candidates.

- On the other hand, it is possible to point out that violent messages come not only from perpetrators of violence who occasionally use social networks for this purpose. Hashtags are often part of a strategy that linked the women's performance and the convention to two major electoral processes: the presidential election in December 2021 and the constitutional plebiscite on 4 September 2022. These two events framed the constitutional debate and were therefore directly related to the type of violence that the women candidates received.

- According to the relevance to certain lists, female candidates may have been more exposed to violence associated with the dispute for the second round of the 2021 presidential election, where José Kast and Gabriel Boric faced each other. Secondly, it is possible to observe that in temporal terms, the discussion and the majority option for the rejection of the new constitutional text was imposed transversally in the hashtag's networks of all the lists of women candidates. It remains to be seen whether it is possible to verify a close link between violence against women on social networks and the extension of the discourse of rejection of the constitutional text. The evidence gathered here seems to indicate this tendency. However, given that this was not the aim of this paper, we believe that further studies are needed to confirm this hypothesis.

- In the particular case of female candidates, their messages highlighting the hopeful character of the process differ profoundly from the animosity shown towards them. The results show that the hashtags used in tweets with hate speeches do not necessarily carry explicit hate messages. Rather, it is discovered that, in a polarized context of strong social crisis, hate speeches circulate strategically associated with hashtags, but without making hateful mentions explicit. No hate speeches were observed in an intersectional perspective that crosses gender, race and even social class, as might have been expected from the tweets.

Among the limitations of this study, it is necessary to note the following. While this is not the only way in which candidates engaged with public opinion, it is striking to note that much of their social networking interactions were informed by the presence of these discourses in the messages and comments they received. It is necessary to consider that there is a lot of cross-hashtag mentions, i.e., that may involve either positive or negative comments about the candidates. There is a lot of criticism that goes through the mention of hashtags. Finally, the very broad category of Independents can be misleading. In this case, the diversity of women's accounts that were grouped in this category may have diminished the diversity of discourses that was expressed during the period.

The results of this study show the close link that other research has detected between hate speech and incivilities. However, what we can conclude from this research is that their use is not only complementary but also strategic. This poses a methodological challenge for future research, since it makes it necessary to articulate the study not only of tweets containing hate, but also their articulation with hashtags and other forms of synthesizing messages for campaign purposes.

This finding may offer an interesting avenue to consider in future research aimed at studying the challenges posed by the increasingly important incorporation of women in positions of political representation. In the context of the current crisis of democracies in several countries, parity policies will be accompanied by a corresponding rise in this type of online violence. It will therefore be increasingly important to understand the logic, the agents and the ends with which this violence is articulated in the context of social networks.

**Author Contributions:** Conceptualization, R.J., and P.W.; methodology, R.J., and J.B.; software, J.B.; validation, J.B., and R.J.; formal analysis, J.B.; investigation, R.J., and P.W.; resources, P.W.; data curation, J.B.; writing—original draft preparation, R.J.; writing—review and editing, R.J., and J.B.; visualization, J.B.; supervision, J.B.; project administration, R.J.; funding acquisition, P.W. All authors have read and agreed to the published version of the manuscript.

**Funding:** This research was funded by United Nations Entity for Gender Equality and the Empowerment of Women (Chile), grant number 103615-SPF 2. O1.1 LEGISLATION.

**Institutional Review Board Statement:** Not applicable.

**Informed Consent Statement:** Not applicable.

**Data Availability Statement:** The databases can be found at the following link and are at the disposal of the journal in case you want to make them public: https://github.com/JarnishsBeltran/AMDoc (accessed on 30 December 2021).

**Conflicts of Interest:** The authors declare no conflict of interest.

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
