# Peer review of "Hate and Incivilities in Hashtags against Women Candidates in Chile (2021–2022)"

_socsci, doi:10.3390/socsci12030180_

Round 1

Reviewer 1 Report

This is a very interesting work and with an appropriate theme in current research.

Most of the doubts refer to the methodology followed:

Methodology

The methodology indicates that quotas and random samples were established for the different analytical exercises carried out, but it does not explain what those quotas were.

It is also specified that a quantitative methodology was used in two phases: the first, based on natural language processing techniques, to detect hate speech and the second, manual coding to label the speeches.

In the first case, it does not detail how this process of downloading and detecting tweets was carried out: if natural language processing techniques were used, what were they? What algorithms were used? What was the process followed? What is the reliability of the detection?

It is indicated that they were based on other studies, but this is not enough, you have to understand the methodology only by reading this article, so there is a lack of information.

In the second phase of manual coding, it remains to specify how this coding was carried out, was a code book used?, with human coders? How many? Were intercoders tested to ensure reliability? If so, what were the results of those tests?

Finally, it indicates that a sentiment analysis was added, but the process that was followed to carry it out is not clear either.

Other details:

It is recommended to improve the visibility of the Figures presented, since being network graphics they are too small to be legible.

Reference 2 is repeated.

Author Response

Dear Reviewer.
We deeply appreciate your review and the comments you have provided.
In the attached document we review one by one and try to respond. Please see the attachment.

Yours sincerely

Reviewer 2 Report

This is a compelling study about gender violence online in Chile. It has great potential and I'll make a few suggestions for improvement so it can be published.

# Theoretical background: The way the central concepts are presented is confusing. It lacks a bit of discussion and connection to the local context. I'd suggest that the authors clearly state what they understand by "incivility", "violence", "hate speech" and other concepts, making clear how these variables were considered by the researchers in this paper. Questions for the authors: Are these concepts all the same? If they are not, which one was the central concept? How was it applied in your research?

#Context: I think Chile is a fascinating context for this research, particularly after Boric's election. However, I know very little about the political spectrum and parties in this country and the local context seems key to understanding the analysis. Since this is an international journal that appeals to an international audience, I'd strongly suggest this political context be better explained. Questions for the authors: What are the parties that appear in your analysis? How do they connect to the left/right spectrum? How the political context in Chile allowed for this new constituency to happen? 

# Methods: It seems to me this paper has a qualitative question and a quantitative method to answer this. While I think there is a strong analysis, it seems that it needs to be better connected to research questions/hypotheses that can create more context for the data. My suggestion would be to present specific research questions with the strategy used to answer them. I'd also advocate for a better discussion on how a network analysis of hashtags can give sufficient context to infer discursive violence (since the interpretation depends on the context of what is said). Questions to authors: How does your analysis effectively infer context from hashtag connection? How each of the metrics of the network analysis are used to give insight into your research questions? 

#Graphs: The graphs could be improved, I couldn't read the concepts on them. Please, make them more visible. 

# Analysis: Because the theoretical background is lacking, the analysis seems superficial. I'd strongly encourage the authors to systematize their findings and show what the key findings are and what one can learn from the study. Questions to authors: How does the analysis connect to your theoretical background? Is there something new that you found and literature doesn't talk about? What are the ways your case study can contribute to understanding other political contexts? 

# Language and format: I'm not sure about the format of some citations. They make weird phrases without a subject. I'd also encourage a general revision of language (as some phrases seem cut) and cohesion between phrases and paragraphs.

Author Response

Dear reviewer. Along with saying hello, I would like to send a detailed response to your review.
Please see the attachment.
Thanks for all your comments
.
Best regards.

Round 2

Reviewer 2 Report

This paper has greatly improved since the last version, and I applaud the authors' efforts. I recommend publication.